

# Use of argon to measure gas exchange in turbulent mountain streams

Robert O. Hall, Jr.[1,3] and Hilary L. Madinger[1,2]

[1]Department of Zoology and Physiology, University of Wyoming, Laramie, WY 82071, USA
[2]Program in Ecology, University of Wyoming, Laramie, WY 82071 USA
[3]Current address: Flathead Lake Biological Station, University of Montana, Polson, MT 59860, USA

**Correspondence:** Robert O. Hall, Jr. (bob.hall@flbs.umt.edu)

**Abstract.** Gas exchange is a parameter needed in stream metabolism and trace gas emissions models. One way to estimate gas exchange is via measuring the decline of added tracer gases such as sulfur hexafluoride ($SF_6$). Estimates of oxygen ($O_2$) gas exchange derived from $SF_6$ additions require scaling via Schmidt number ($Sc$) ratio, but this scaling is uncertain under conditions of high gas exchange via bubbles because scaling depends on gas solubility as well as $Sc$. Because argon (Ar) and

$O_2$ have nearly identical Schmidt numbers and solubility, Ar may be a useful tracer gas for estimating stream $O_2$ exchange. Here we compared rates of gas exchange measured via Ar and $SF_6$ for turbulent mountain streams in Wyoming USA. We measured Ar as the ratio of $Ar:N_2$ using a membrane inlet mass spectrometer. Normalizing to $N_2$ confers higher precision than simply measuring [Ar] alone. We consistently enriched streams with Ar from 1% to 15% of ambient Ar concentration and could estimate gas exchange using an exponential decline model. The mean ratio of gas exchange of Ar relative to $SF_6$ was

1.8 (credible interval 1.1 to 2.5) compared to the theoretical estimate 1.35, showing that using $SF_6$ would have underestimated exchange of Ar. Steep streams (slopes 11-12%) had high rates of gas exchange velocity normalized to $Sc = 600$ ($k600$, 57-210 m d$^{-1}$), and slope strongly predicted variation in $k600$ among all streams. We suggest that Ar is a useful tracer because it is easily measured, requires no scaling assumptions to estimate rates of $O_2$ exchange, and is not an intense greenhouse gas as is $SF_6$. We caution that scaling from rates of Ar or $SF_6$ gas exchange to $CO_2$ is uncertain due to solubility effects in conditions

of bubble mediated gas transfer.

## 1   Introduction

Air-water gas flux is a key process in aquatic ecosystems because it defines the flow of material between water and the atmosphere. Knowing this flux is needed for questions ranging from global $CO_2$ balance (Raymond et al., 2013) to short term $O_2$ budgets to estimate ecosystem metabolism (Odum, 1956). Gas flux is the product of air-water gas exchange velocity ($k$, m d$^{-1}$)

and the relative saturation in water, i.e., $F = k(\alpha C_{air} - C_{water})$ where $C_{air}$ and $C_{water}$ are the concentrations of gas in the air and water and $\alpha$ is the unitless Ostwald solubility coefficient. The gas exchange velocity, $k$ (m d$^{-1}$), is a central variable for estimating gas flux, and it is much harder to measure than the air-water concentration gradient in gases. $k$ can vary greatly through time and space and thus requires many empirical measurements or robust predictive models to accurately estimate gas exchange.



There are several ways to measure gas exchange in aquatic ecosystems. In places with high rates of primary production and low gas exchange, it is possible to measure gas exchange rates via diel curves of oxygen with time (Hornberger and Kelly, 1975; Holtgrieve et al., 2010; Hall et al., 2016; Appling et al., in press). Direct measures with domes are practical in low-exchange habitats (Borges et al., 2004; Alin et al., 2011). Tracer gas additions is another effective means of measuring gas

exchange across all types of aquatic habitats (Wanninkhof et al., 1990; Wanninkhof, 1992; Cole and Caraco, 1998). Tracer additions are particularly useful because they represent direct measures at spatial scales similar to that of turnover length of gases. Given enough estimates of $k$, it is then possible to build theory of gas exchange across time and space (Wanninkhof, 1992; Raymond et al., 2012), e.g., among small high energy streams. A trade off with gas exchange measured by tracer gases is that it is necessary to scale exchange rates measured for the tracer gas (e.g., $SF_6$, propane, $^3He$) with that of gases of ecological

interest (e.g., $O_2$, $CO_2$, $CH_4$). This scaling is not always straightforward because high rates of bubble mediated gas exchange cause scaling to depend on differences in solubility of gases as well as their diffusivity (Asher and Wanninkhof, 1998a, b; Woolf et al., 2007). Thus, an ideal tracer gas would not require scaling if its solubility and diffusivity were similar to the gas of ecological interest. Here we demonstrate the use of argon (Ar) as a tracer gas; Ar has similar solubility and diffusivity to $O_2$, a gas of major biological interest in the context of estimating metabolism in aquatic ecosystems (Odum, 1956; Nicholson et al.,

2015; Bernhardt et al., 2017).

In the absence of extensive bubbles, one can scale gas exchange rates between gases based on the ratio of their Schmidt numbers ($Sc$); $Sc$ is the dimensionless ratio of kinematic viscosity of water ($\nu$) and the diffusion coefficient of the gas ($D$), i.e., $Sc = \frac{\nu}{D}$. Given $Sc$ for two gases, scaling gas exchange rates is given by

$$\frac{k_1}{k_2} = \left(\frac{Sc_1}{Sc_2}\right)^{-n} \tag{1}$$

(Jähne et al., 1987) where $-n$ is a coefficient ranging from 0.67 for smooth water to 0.5 for wavy water. When bubbles are present scaling between gases depends upon solubility of the gases in addition to their diffusivity (Asher and Wanninkhof, 1998b). This bubble effect $k_b$ is additive to that of an unbroken surface ($k_o$) such that $k_w = k_o + k_b$ (Goddijn Murphy et al., 2016). One model for the bubble-mediated component of gas exchange, $k_b$ is given by eq 13 in Woolf et al. (2007):

$$k_b = \frac{Q_b}{\alpha} \times (1 + \chi^{1/f})^{-f}$$
$$\chi = \frac{Sc^{0.5}}{14\alpha} \tag{2}$$

where $Q_b$ is the bubble flux and $f = 1.2$. We can compare the ratios of the bubble-mediated component gas exchange $k_{b,1}/k_{b,2}$ for 2 gases with varying solubility $\alpha_1$ and $\alpha_2$ as

$$\frac{k_{b,1}}{k_{b,2}} = \frac{\alpha_2}{\alpha_1} \times \left(\frac{1 + \chi_1^{1/f}}{1 + \chi_2^{1/f}}\right)^{-f} \tag{3}$$

This model shows that the effect of varying solubility on scaling $k_b$ among gases depends on the solubility (Fig 1). For low-solubility gases such as Ar and $SF_6$, this model predicts only a Schmidt number effect. For more soluble gases, such as $CO_2$,





the scaling factor is higher than what would be predicted because of the higher solubility of $CO_2$ (Fig 1). Here, we test the gas exchange scaling of two sparingly soluble gases, Ar and $SF_6$ in high energy mountain streams with presumably high rates of bubble-mediated gas exchange.

Argon is promising for measuring gas exchange because it has low background concentrations in water, it is inert, it is cheaply available from welding supply stores, it has similar solubility and diffusivity to $O_2$ (Fig 1), and it is easily detected using membrane inlet mass spectrometry. We compared Ar to $SF_6$, another commonly used tracer gas that supersedes Ar in detectability, but has higher Schmidt number and lower solubility in addition to being an intense greenhouse gas. Our objectives were:

1. Develop a method to measure gas exchange in streams using Ar tracer additions.

2. Test scaling of Ar to $SF_6$ in turbulent streams with high rates of bubble mediated gas transfer.

## 2 Methods

### 2.1 Sites

We sampled five streams across a gradient of predicted gas exchanges to compare performance of Ar and $SF_6$ as tracers. Streams were headwaters in Southeast Wyoming ranging from three mountain streams in Snowy and Laramie Ranges (NoName Creek, Pole Creek, and Gold Run), one urban spring stream (Spring Creek), and a low-slope, meadow stream in Vedauwoo area of the Laramie Range (Blair Creek) (Table 1). The three mountain streams were steep channels with step-pool morphology and presumed high rates of gas exchange.

### 2.2 Gases and NaCl injection

We added Ar and $SF_6$ gases to each stream and modeled their downstream evasion to estimate their relative exchange rates. Prior to injection, we collected pre-plateau samples at each of six sampling locations and an upstream location. We collected dissolved $Ar:N_2$ samples using a 3.8 cm diameter PVC pipe with an attached outlet tube (3.2 mm ID × 20 cm vinyl tube) at the downstream end. As water flowed through the pipe, we capped the downstream end with a stopper. Lifting from the stream, water flowed through outlet tube to >triple overflow a 12 mL Exetainer that we capped immediately without bubbles. We did not use preservative because we analyzed samples within a week and we found no change in concentration of these nearly inert gases in this time period using laboratory tests. We measured specific conductivity using a hand held conductivity sensor or conductivity and temperature using a Onset HOBO conductivity logger and converted the values to specific conductivity at each sampling location. We also recorded the stream temperature using a reference Thermopen (ThermoWorks, American Fork, UT) and barometric pressure in mmHg using a hand held barometer (Exetech, Nashua, NH) to calculate saturated dissolved gas concentrations. We assumed $SF_6$ concentration was 0 before the addition.

Following pre-injection sampling, we simultaneously injected Ar, $SF_6$, and a NaCl solution. We bubbled Ar using a micro bubble ceramic diffuser (Point Four Systems Inc., Coquitlam, BC) from a compressed Ar tank at a constant bubbling rate ~0.2



m$^3$ h$^{-1}$. SF$_6$ was bubbled at $\sim$100 mL min$^{-1}$ through a needle valve attached to a variable area flow meter and to a 30 cm aquarium air stone. Concurrently we injected a NaCl solution at a constant rate using a peristaltic pump. Salt solution flow rates were enough to increase stream conductivity by 20 to 50 $\mu$S cm$^{-1}$. Once the downstream station reached plateau conductivity, we sampled each station for specific conductivity, stream temperature, barometric pressure, and triplicate dissolved

gas concentration as for the pre-injection sampling. Additionally, we sampled SF$_6$ by sucking 45 mL of stream water into a 60 mL plastic syringe and adding 15 mL of air. The syringe was shut using a stopcock and shaken for 5 minutes. The 15 mL of headspace were injected into an evacuated 12 mL Exetainer. We collected 3 SF$_6$ samples at each station. We collected all samples in an upstream to downstream sequence and we stored these samples cooler than stream temperature to prevent out gassing.

We measured stream physical variables. We estimated stream discharge, $Q$ based on dilution of the NaCl tracer. Nominal transport time ($t$) was estimated as the time to reach 1/2 of the plateau concentration of conductivity. Stream velocity ($v$) was reach length, measured by meter tape, divided by $t$. We measured the stream mean wetted width at >8 locations at constant intervals through the sampling reach.

### 2.3 Ar and N$_2$ analysis

We measured dissolved Ar:N$_2$ in water samples using a membrane inlet mass spectrometer (MIMS) (Bay Instruments Inc., Easton, MD) (Kana et al., 1994). We used a two point calibration by setting water bath temperatures $\pm 2°$C the sample collection temperature. Round-bottom flasks in each water bath equilibrated with the atmosphere by stirring at $\sim$200 rpm. We bracketed groups of 5-10 samples with calibration samples from each water bath. We recorded the currents at $m/z$ 28 and 40, and their ratio from the mass spectrometer (Kana et al., 1994).

We converted the ratio currents $m/z\ 40 : m/z\ 28$ to Ar:N$_2$ ratios. We normalized all Ar measures to N$_2$ because the MIMS is more precise with gas ratios than absolute concentrations. We calculated the Ar:N$_2$ in each of the two standard flasks assuming that they were in equilibrium with the atmosphere at a known temperature and barometric pressure. We estimated saturation concentrations in each flask based on Hamme and Emerson (2004). Unknown Ar:N$_2$ in each sample was calibrated using a linear relationship derived from the Ar:N$_2$ in the 2 standard flasks. Despite adding Ar to the streams, the amount of Ar was

not high relative to ambient Ar. Based on the small enrichment of Ar, we assumed N$_2$ concentration changed little during the injection and through the reach due to displacement by Ar in the reach because denitrification causes a uniform and small increase to the N$_2$ concentration compared to saturation throughout the reach.

### 2.4 SF$_6$ analysis

We measured SF$_6$ at the Utah State University Aquatic Biogeochemistry Lab using a gas chromatograph (GC) (SRI Instru-

ments, Torrance, CA) with an electron capture detector. We injected 5-20 $\mu$L of samples into the GC for analysis. From each measurement, we estimated the relative SF$_6$ concentration as area of the peak divided by injection volume. We assumed no SF$_6$ present in streams naturally and therefore use a pre-plateau value of 0. Blanks showed no SF$_6$.





## 2.5   Data analysis and inference

We estimated gas exchange rates assuming a first-order decay with distance. Let $A$ represent the excess Ar:N$_2$ and $S$ excess SF$_6$ (measured as peak area $\times$ injection volume) in stream water corrected for groundwater inputs. $C$ is specific electrical conductivity ($\mu$S cm$^{-1}$) First, at each site, $x$ we estimated a groundwater-corrected $A_x$ and $S_x$ as

$$A_x = \frac{[Ar:N_2]_{x,plateau} - [Ar:N_2]_{x,ambient}}{C_{x,plateau} - C_{x,ambient}}$$

$$S_x = \frac{[SF_6]_{x,plateau}}{C_{x,plateau} - C_{x,ambient}}.$$

(4)

where $plateau$ and $ambient$ indicate samples collected during and before the gas and salt additions. We estimated ambient Ar based upon temperature at each site during the collection time of the plateau samples. Measured ambient Ar:N$_2$ accurately matched the calculated ambient but had higher within-site variability due to measurement error, thus we assumed that ambient Ar:N$_2$ was that estimated based on saturation calculations (Supplemental material). We normalized $A_x$ and $S_x$ to that of their upstream-most concentrations, i.e., at the first sampling station below the injection ($A_0$, $S_0$)

$$An_x = \frac{A_x}{A_0}, Sn_x = \frac{S_x}{S_0},$$

(5)

We fit exponential decay statistical models to the data

$$An_x \sim \mathcal{N}(An_0 \times e^{-Kd \times x}, \sigma_A)$$

$$Sn_x \sim \mathcal{N}(Sn_0 \times e^{\frac{-Kd}{a} \times x}, \sigma_S)$$

(6)

where $Kd$ is the per length evasion rate of Ar, and $a$ is the ratio of exchange rates between Ar and SF$_6$. This model assumes that both Ar and SF$_6$ evaded as a exponential function of distance downstream, $x$, and that residual errors are normally distributed with a mean of 0 and standard deviations $\sigma_A$ and $\sigma_S$ for SF$_6$, Parameters in this model are $An_0$, $Sn_0$, $Kd$, $a$, $\sigma_A$, and $\sigma_S$.

We fit these models within a hierarchical Bayesian framework. We were most interested in the value of $a$, i.e., the ratio of gas exchange for Ar and SF$_6$. For any stream, $j$, we estimated $a_j$ by using partial pooling across additions such that its prior probability was

$$a_j \sim \mathcal{N}(a_{mean}, \sigma_a)$$

(7)

where $a_{mean}$ and $\sigma_a$ are themselves distributed as $a_{mean} \sim \mathcal{N}(1.36, 1)$. This normal prior distribution had a mean of 1.36, which is the expected ratio of $k_{Ar}$:$k_{SF6}$ based on Eq. 1, and a standard deviation of 1 allowing for considerable variation in $a_{mean}$ from 1.36. The among stream variation $a_j$ ($\sigma_a$) had a half-normal prior distribution of $\sigma_a \sim |N(0,2)|$. Prior probability for $Kd$ was $Kd \sim \mathcal{N}(0, 0.1)$ where -0.1 would be a very high rate of gas exchange. Prior probabilities for $An_0$ and $Sn_0$ were $Kd \sim \mathcal{N}(1, 0.05)$.





We fit this model using by simulating the posterior parameter distributions using the program Stan (Stan Development Team, 2016) using the rstan package in program R (R Core Team, 2016). Stan uses a Markov-chain Monte Carlo (MCMC) method to simulate posteriors. For each parameter we ran 4 MCMC chains with 500 steps burn in and 1000 for sampling. We visually checked the chains for convergence and that the scale reduction factor, $\hat{R} < 1.1$ for all parameters.

We converted the per distance rate to gas exchange of Ar to per unit time ($K$, d$^{-1}$) as $K = Kd/v$ where $v$ is stream velocity (m d$^{-1}$). Gas exchange velocity ($k$, m d$^{-1}$) was calculated as

$$k = Q \times \frac{kd}{w} \tag{8}$$

To facilitate comparison with other studies, we scaled our temperature specific estimates of $k$ from each stream to $k$ at a Schmidt number of 600 ($k600$) following Eq. 1 using equations to estimate $Sc$ from Raymond et al. (2012).

## 3   Results

We enriched all streams with Ar and estimated gas exchange rates with varying precision. Enriched Ar:N$_2$ at the first station downstream from the addition site averaged 7% higher than the ambient Ar:N$_2$ (range 1.2 to 15.6%). This low enrichment was large enough to easily measure a decline in Ar:N$_2$ to ambient (Fig. 2), but low enough to minimally affect absolute N$_2$ concentration via degassing of N$_2$ if we had e.g., enriched Ar 10-fold. Gas exchange rates, $Kd$ ranged from 0.00067 to 0.050

m$^{-1}$ and 95% credible interval on these rates averaged 0.42 % (range 36-54%) of the rate itself. Precision on our Ar:N$_2$ measures was high. The median standard deviation of replicate samples at each station was $3.31 \times 10^{-5}$, corresponding to a coefficient of variation (cv) of 0.09%. The cv for Ar conc was $2.5\times$ higher at 0.23% showing that normalizing Ar by N yielded more precise estimates. Coefficient of variation for replicates of SF$_6$ analyses was 5%, much higher than that for Ar:N2.

    Ratios of $K_{Ar} : K_{SF6}$ measured in each injection varied greatly and were higher than the expected ratio of 1.36. These

ratios ($a_j$) varied from 0.6 to 3.4 (Table 1) and the mean of the pooled ratio ($a_{mean}$) was 1.8 with a 95% credible interval, 1.1-2.5. Variation among releases was high, with $\sigma_a = 0.9$. The credible interval in $a$ averaged 49% of $a$ showing that estimates of SF$_6$ evasion had slightly more uncertainty than that for $Kd$. This finding is despite the fact that $\sigma_S$ was lower than $\sigma_A$, likely because some values of normalized $A$ ($An_x$) were negative. Negative values of $An_x$ increase $\sigma_S$, but do not necessarily increase uncertainty in the estimate of $Kd$.

Variability in $a$ led to potential for error in estimating $k600$ between Ar and SF$_6$. $K600$ based on SF$_6$ was lower than that for Ar for 6 of the 8 additions (Fig. 3). Deviance from a 1:1 line exceeded that of the statistical errors around $Kd$ in the models because the posterior distributions themselves deviated from the 1:1 line (Fig. 3).

    Gas exchange was high at our steep streams. Gas exchange velocity ($k600$) ranged from 5.4 to 208 m d$^{-1}$, and covaried tightly with variation in stream slope (Fig. 4). The $k600$ from our streams were much higher than most literature values; the 4

sites with slopes $\geq 0.05$ exceeded 99% of the values in Raymond et al. (2012). The per-time rate of gas exchange ranged from 28 to 740 d$^{-1}$ (Table 1).





## 4 Discussion

Despite low enrichment of Ar:N$_2$, we estimated $Kd$ based upon exponential declines of this tracer gas signal. On the surface, one might consider Ar to be a poor tracer gas because it is the third most abundant gas in the atmosphere at 1% concentration, thus requiring a large increase in concentration to detect a decline. But because MIMS is highly precise when measuring gas

ratios (Kana et al., 1994), it is not necessary to elevate concentrations greatly. This low enrichment has two advantages. One is the practical aspect of not needing to haul a big tank of gas to a remote stream (a 2.2-kg tank lasted us for several additions). The second is that the Ar bubbling stripped little of the N$_2$ from the stream, thus we did not need to model the concomitant invasion of N$_2$. That said, there needs to be enough Ar to have a high signal to noise ratio to detect a decline in Ar. We suggest at least a 3% increase in the Ar concentration. Given that measurement error with the MIMS is constant across a range

of concentrations, all things equal, higher values of Ar:N$_2$ are better. We did not test the conditions under which we could increase the incorporation rate of Ar into streams, but it seems reasonable to assume that higher Ar flow rate, larger air stones and deeper pools in which to inject Ar would all increase values of Ar:N$_2$. We used a fine-bubble air stone and suggest that this device greatly facilitated Ar exchange. One needs to be aware of changing temperature between the ambient and during plateau samples. Changing temperature 5°C can cause a 1% change in Ar:N$_2$; hence, one needs to estimate ambient Ar:N$_2$

during plateau if temperature is changing either by calculating ambient Ar:N$_2$ at sampling temperature or monitoring at an upstream station.

Our estimates of the ratio of $Kd_A : Kd_S$ ($a$) were higher than the 1.36 expected based on Schmidt number scaling (Jähne et al., 1987) and the 1.33 based on Eq 3. This ratio $a$ also varied greatly among injections, such that we had high uncertainty on the actual value of $a$ (Table 1). Thus, there are two problems. One is estimates of $Kd$ for either tracer gas contained substantial

error leading to high variation in estimates of $a$. The other is that $a$ was inexplicably higher than predicted for both smooth surface and bubble-mediated transfer. Either the theory for scaling in equations 1 and 3 (Woolf et al., 2007; Goddijn Murphy et al., 2016) did not work in our case or we estimated either $Kd_A$ or $Kd_S$ with bias. From a theoretical perspective, this question behind $a > 1.36$ is compelling, because if true it complicates models of bubble-mediated gas exchange (Goddijn Murphy et al., 2016). From a practical perspective—where one simply needs to estimate $k600$ for O$_2$ exchange—this question is less germane

given that one could simply use Ar rather than SF$_6$. If one uses tracer estimates for SF$_6$, and our estimate of $a$ was in fact 1.8, then all else equal, gas exchange will be estimated at $1.36/1.8 = 0.75$ lower than the true value, which we observed for 6 of the 8 injections (Fig. 3). If using these gas exchange rates to estimate metabolism, then estimates of ecosystem respiration will also be $0.75\times$ too low. This bias in ecosystem respiration may be small relative to the effects of groundwater, probe calibration, and process error (Appling et al., in press), but this bias adds to the complications in estimating ecosystem respiration from

diel O$_2$ data (Demars et al., 2015).

In steep, turbulent streams and rivers, bubbles likely cause most of the gas exchange (Hall et al., 2012), complicating scaling among gases because one must consider variation in solubility as well as variation in $Sc$. Theory from Woolf et al. (2007) shows that at low solubilities variation in $Sc$ is all that is needed to scale among gases. Thus scaling from SF$_6$ to Ar or O$_2$ may be constant as $k_b$ approaches $K_w$. Although we did not assess propane in this study, based on the similarity of propane $Sc$ and

$\alpha$ with $SF_6$, it is likely that there is not a strong solubility effect on its rate of $k_b$. For gases with much higher solubility, i.e., $CO_2$, scaling may deviate strongly when bubbles dominate gas exchange (Fig. 1) because bubbles do not reach equilibrium and this scaling depends upon both $Sc$ and solubility. Such streams have high rates of gas exchange and error in estimating $k$ for $CO_2$ that may greatly affect flux estimates in these streams. Thus we caution using the findings here for estimating $CO_2$ flux in streams with high turbulence. In addition, our subsequent work (A. J. Ulseth et al. unpublished data) will show that it is not possible to predict $k600$ in highly turbulent stream based on models from low energy streams and rivers (Raymond et al., 2012). Streams with steep slopes, such as our 4 steepest streams, have much higher gas exchange than would be predicted from current empirical models (Raymond et al., 2012).

## 5   Conclusions

We recommend using Ar as a tracer gas in small streams. Argon is an inert and easily obtained gas that one can precisely measure using MIMS. In addition, Ar is not a greenhouse gas. While $SF_6$ is inert and easily detectable, thus making a potentially ideal tracer, $SF_6$ has $23,500\times$ the greenhouse forcing of $CO_2$ (Myhre et al., 2013). It is somewhat ironic to study carbon cycling with a tracer gas with that much greenhouse forcing. If one is interested in $O_2$ exchange, then Ar is an optimal tracer because it has nearly the same solubility and diffusivity of $O_2$, thus eliminating the need to scale between gases. Given uncertainty with scaling due to bubbles and the higher than predicted scaling ratio ($a$) found here, scaling from $SF_6$ to $O_2$ is somewhat uncertain. $SF_6$ does hold the advantage as a gas tracer for large streams and rivers. We focused only on small streams here and have not tested this method on larger streams and rivers. One would need to add much more Ar, which is difficult, but possible with larger tanks and air stones. $SF_6$ is so detectable that it is used in very large rivers (Ho et al., 2011). But it may be easier to measure gas exchange in large rivers using diel cycling of $O_2$ in lieu of a tracer (Hall et al., 2016). In fact, with low gas exchange, diel $O_2$ cycling may provide more accurate estimates of $k600$ than tracer additions that extend for multiple km downstream (Holtgrieve et al., 2015) and with a long time series of diel $O_2$, one can obtain even better estimates of $k600$ (Appling et al., in press). The Ar method we present here, however, worked well in small, steep streams where high rates of gas exchange required empirical measurements for accurate estimates of $k600$.

*Code and data availability.*   Code and data for all analyses are available as supplementary materials

*Author contributions.*   ROH and HLM designed the study, conducted fieldwork, measured $SF_6$, and analyzed data. HLM measured Ar:$N_2$. ROH wrote the first draft of the paper and made the figures.

*Competing interests.*   The authors declare no competing interests.





*Acknowledgements.* Ina Goodman, Alison Appling, Pavel Garcia, Keli Goodman, Brady Kohler, Brittany Nordberg, and Rachel Usher assisted with fieldwork. Michelle Baker and Autumn Slade set us up on their GC and provided food. Financial support came from National Science foundation grants and EF-1442501. Amber Ulseth, Tom Battin, and Lauren Koenig read and commented on early drafts of this paper. We dedicate this paper to the memory of Ina Goodman.





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





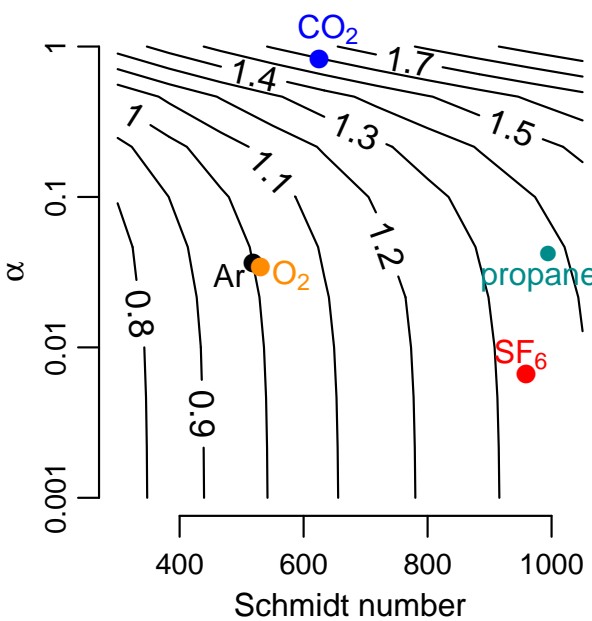

**Figure 1.** Bubble mediated gas exchange rate ($K_b$) normalized to that of argon. Temperature was at 20°C. Contours are equivalent to $K_{b,Ar}/K_{b,2}$ where $K_{b,2}$ varies as a function of solubility and Schmidt number. At low solubilities ($\alpha$, Ostwald solubility coefficient), scaling among gases depends only on variation of Schmidt number (SF$_6$). As solubility increases, scaling depends on both Schmidt number and $\alpha$ (CO$_2$). O$_2$ is similar to Ar. Propane has similar properties to SF$_6$. Analysis based on Eq. 13 in Woolf et al. (2007).





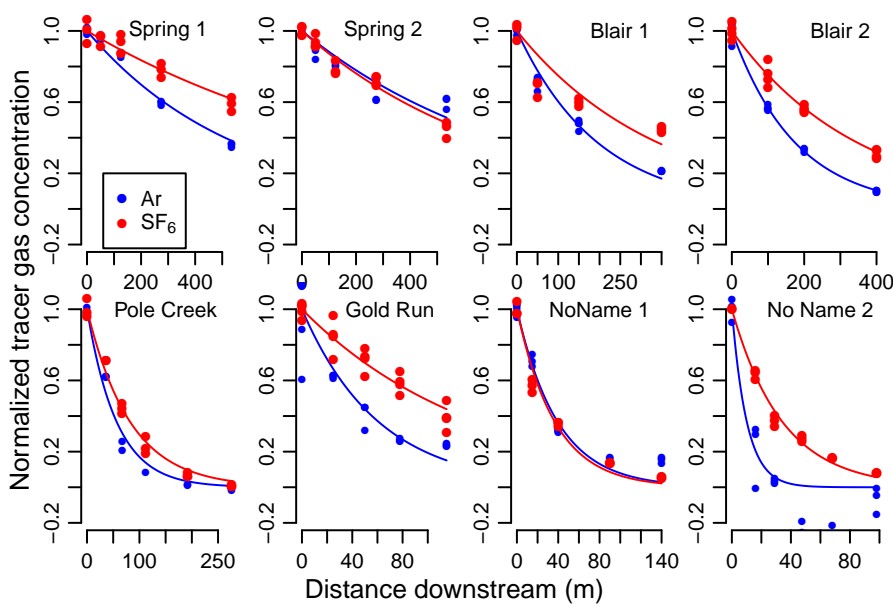

**Figure 2.** Exponential decline of normalized argon and SF$_6$ at each downstream sampling site for each stream shows that rates of decline for SF$_6$ (blue) are lower than that for Ar (red). Points are normalized tracer gas concentrations, $An_x$ and $Sn_x$, and lines are exponential model fits (Eq. 6).





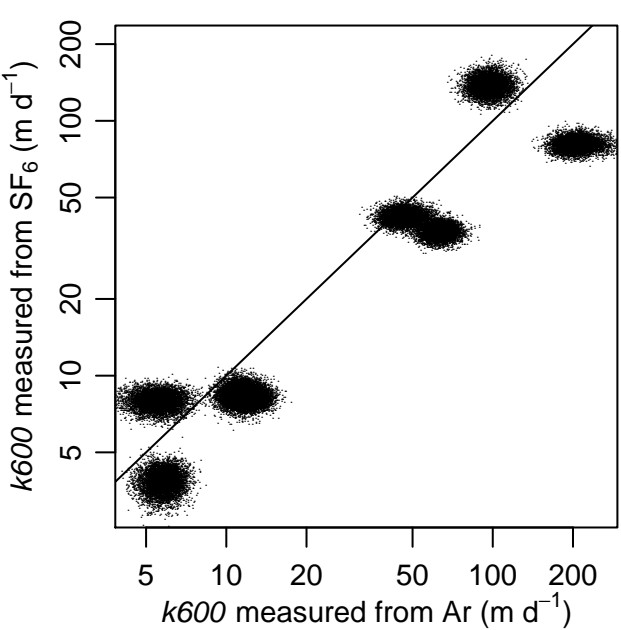

**Figure 3.** $k600$ measured from $SF_6$ was lower than predicted from $k600$ measured from argon in 6 of the 8 injections. Each injection is represented by a cloud of points that represents 6000 draws from the posterior distributions of $k_j$ and $a_j$, from which we calculated gas exchange velocity ($k$) following Eq. 8 and converted to $k600$ using Eq 1. Line is 1:1. Note log-scaled axes.





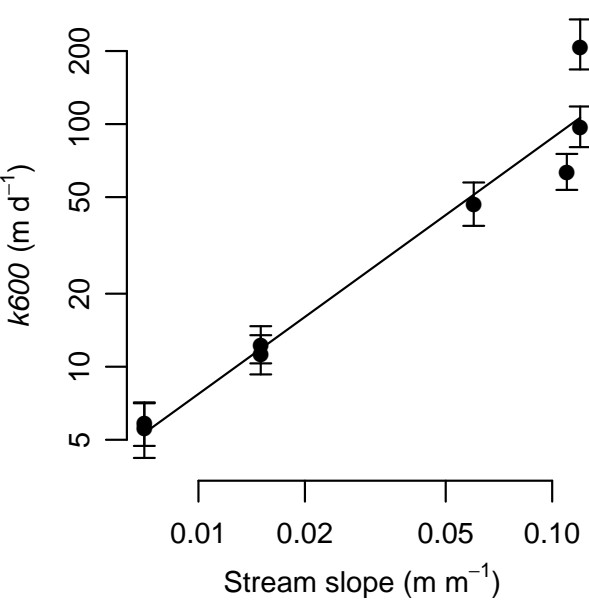

**Figure 4.** Gas exchange velocity increased as a power relationship with stream slope. Equation is $ln(k600) = 1.055 \times ln(slope) + 6.90$. Error bars are 95% credible intervals.





**Table 1.** Site data for streams sampled including average stream width, reach length, solute travel time ($t$, water velocity $v$, discharge ($Q$), average stream depth ($\bar{z}$), stream slope, and average stream temperature during plateau. Gas exchange parameters $k_{Ar600}$ and $a$ are reported with 95% credible intervals. We also include the per time rate of gas exchange, $K_{60}$ (d$^{-1}$)

| Site | Date | Width (m) | Reach length (m) | Travel time (min) | Velocity (m min$^{-1}$) | $Q$ (m$^3$/s) | $\bar{z}$ (m) | Slope (m/m) | Temp. (°C) | $K_{600}$ (d$^{-1}$) | $k_{Ar600}$ (m d$^{-1}$) | $a$ |
|---|---|---|---|---|---|---|---|---|---|---|---|---|
| Spring 1 | 13-Aug-15 | 2.3 | 300 | 25 | 12 | 0.084 | 0.18 | 0.007 | 17.4 | 31 | 5.8 (4.7,7.1) | 2.0 (1.5,2.6) |
| Spring 2 | 30-Jun-16 | 1.6 | 860 | 56 | 15.4 | 0.070 | 0.17 | 0.007 | 12.6 | 28 | 5.6 (4.2,7.2) | 0.92 (0.67,1.2) |
| Blair 1 | 2-Jul-15 | 0.8 | 420 | 44 | 9.5 | 0.020 | 0.16 | 0.015 | 17.5 | 69 | 11 (9.3,14) | (1.4,2.2) |
| Blair 2 | 15-Jul-15 | 0.8 | 420 | 44 | 9.5 | 0.020 | 0.16 | 0.015 | 18.2 | 77 | 12 (10,15) | 2.0 (1.6,2.4) |
| Pole | 12-May-17 | 0.9 | 300 | 98 | 3.1 | 0.021 | 0.45 | 0.05 | 9.4 | 79 | 47 (38,58) | 1.5 (1.2,1.9) |
| Gold Run | 11-Oct-16 | 3.3 | 140 | 35 | 4 | 0.097 | 0.44 | 0.113 | 4.4 | 95 | 64 (54,75) | 2.3 (1.9,2.8) |
| NoName 1 | 26-Jun-15 | 1.3 | 233 | 37 | 6.3 | 0.022 | 0.30 | 0.12 | 6.7 | 230 | 97 (80,120) | 0.93 (0.73,1.2) |
| NoName 2 | 14-Jul-16 | 0.7 | 110 | 21 | 5.2 | 0.022 | 0.20 | 0.12 | 6.3 | 740 | 208 (170,270) | 3.4 (2.7,4.5) |