# Peer review of "Use of argon to measure gas exchange in turbulent mountain streams"

_Biogeosciences, 2018_

## Referee Comment (RC1) · Anonymous Referee #1 · 29 Mar 2018

This is a well produced manuscript that introduces an important and new methodological approach to measuring reaeration in high-gradient streams. The study was well executed and comprehensive. Because the methods and results are relatively straightforward and the design of the study relatively simple, I have no further comments for improvement.

---

## Referee Comment (RC2) · D. F. McGinnis (Referee) · 4 Apr 2018

"Use of argon to measure gas exchange in turbulent mountain streams", by Robert O. Hall and Hilary L. Madinger. In their manuscript, the authors use Argon and SF6 to determine the gas exchange rate in streams of varying slopes. The manuscript it straight forward, concise and convincing, and the topic is certainly timely. The manuscript is a very nice contribution.

My only concern is the effect of the introduced bubbles on the dissolved N2 concentrations. Since the authors use the Ar/N2 ratio, this may have consequences for their calculations. I performed some bubble simulations on shallow streams using various bubbles (pure Ar, O2, etc) and according to the results, a 1 mm diameter Ar bubble will

strip as much N2 out of the water as the Ar that is dissolved. If this has a relevant impact on dissolve N2 concentrations, then it would translate to artificially high K values using the Ar/N2 ratio and might explain their higher reported ratio of gas exchange of Ar to SF6. Perhaps the authors can provide additional information if the N2 stripping by bubble addition is truly negligible for their k calculations? A simple test with measuring N2 (or even O2 as it should scale) immediately upstream and downstream of their bubble addition would be compelling.

Minor comment: Pg 4, line 25. Last sentence of that paragraph is a bit unclear.

---

## Author Comment (AC1) · 10 Apr 2018

xcolor

**Response to reviews, Hall and Madinger "Use of argon..."**

Reviewer text in black

Author response in blue

**Reviewer 1** This is a well produced manuscript that introduces an important and new methodological approach to measuring reaeration in high-gradient streams. The study was well executed and comprehensive. Because the methods and results are

relatively straight- forward and the design of the study relatively simple, I have no further comments for improvement.

Thank you.

**Reviewer 2, D. McGinnis** "Use of argon to measure gas exchange in turbulent mountain streams", by Robert O. Hall and Hilary L. Madinger. In their manuscript, the authors use Argon and SF6 to determine the gas exchange rate in streams of varying slopes. The manuscript it straight forward, concise and convincing, and the topic is certainly timely. The manuscript is a very nice contribution.

My only concern is the effect of the introduced bubbles on the dissolved N2 concentrations. Since the authors use the Ar/N2 ratio, this may have consequences for their calculations. I performed some bubble simulations on shallow streams using various bubbles (pure Ar, O2, etc) and according to the results, a 1 mm diameter Ar bubble will strip as much N2 out of the water as the Ar that is dissolved. If this has a relevant impact on dissolved N2 concentrations, then it would translate to artificially high K values using the Ar/N2 ratio and might explain their higher reported ratio of gas exchange of Ar to SF6. Perhaps the authors can provide additional information if the N2 stripping by bubble addition is truly negligible for their k calculations? A simple test with measuring N2 (or even O2 as it should scale) immediately upstream and downstream of their bubble addition would be compelling.

We completely agree with the reviewer that Ar will strip some of the $N_2$ from the stream. We briefly mentioned this point in the paper, and we now have added text to the discussion to quantify the worst case scenario of how much $N_2$ was stripped out. The short answer is very little because we added small quantities of Ar and the Ar pool is much smaller than the $N_2$ pool . Indeed the amount of $N_2$ stripped would be so small that it would be difficult to measure using MIMS to estimate concentrations, as opposed to ratios.

Here is a an example. We enriched Ar from 1.01 to 1.18 of its ambient concentration. Choosing the high value of 1.18 corresponds to in an increase in the dissolved Ar from 0.476 mg L$^{-1}$ to 0.561 mg L$^{-1}$. This difference corresponds to an enrichment of 0.00214 mmol Ar L$^{-1}$. Assuming a mole for mole exchange with $N_2$ gas, there would be a 0.00214 mmol L$^{-1}$ decline in $N_2$ from its saturation concentration of 0.455 mmol L$^{-1}$. This value represents a 0.47% decline in dissolved $N_2$. This decline would not be measurable if we measured $N_2$ concentrations alone (the MIMS is much more precise with ratios than concentrations). Our standard deviation of replicate measures of $N_2$ is 0.6% of saturation concentration.

We do recognize that going gung-ho with a huge tank of Ar and big diffusing stones could enrich the stream to e.g., 10× the Ar concentration. Such enrichment would cause about a 23% decline in $N_2$. In this situation, one would need to model the $N_2$ invasion.

The results section includes the level of enrichment of Ar above it saturation concentration, but we will clarify the discussion to explicitly decline the potential $N_2$ stripping. We will add the following text to the discussion:

" A potential concern when conducting these experiments is excess Ar bubbled to the stream will strip $N_2$ as Ar diffuses from bubbles and $N_2$ diffuses in. If this $N_2$ flux is large, one would need to model the concomitant invasion of $N_2$ as well as the evasion of Ar. How much $N_2$ did the Ar strip? We averaged an enrichment of 7% of ambient Ar concentration with a high of 18%. This high value corresponds to in an increase in dissolved Ar from 0.476 mg L$^{-1}$ to 0.561 mg L$^{-1}$, which is an enrichment of 0.00214 mmol Ar L$^{-1}$. Assuming a mole for mole exchange with $N_2$ gas, there would be a 0.00214 mmol L$^{-1}$ decline in $N_2$ from its saturation concentration of 0.455 mmol L$^{-1}$. This value represents a 0.47% decline in dissolved $N_2$, a small amount relative to the 18% increase in Ar."

Minor comment: Pg 4, line 25. Last sentence of that paragraph is a bit unclear.

We will revise this long sentence into 3 shorter ones "Based on the small enrichment of Ar, we assumed that $N_2$ concentration changed little during the injection due to bubble exchange with Ar. In addition we assumed no biologically driven $N_2$ fluxes. Denitrification would cause a uniform and small increase to the $N_2$ concentration compared to saturation throughout the reach."

---

## Referee Comment (RC3) · D. F. McGinnis (Referee) · 17 Apr 2018

Thank you for your response to my previous comment. I agree with your argumentation that in this case, the influence on the N2 should be negligible, and this is now reflected in the text.

---

## Author Comment (AC2) · 18 Apr 2018

We thank Dr. McGinnis for his comments. We will revise the manuscript as we detailed in our previous author response comment.